# Economical Chemical Recycling of Complex PET Waste in the Form of Active Packaging Material

**DOI:** 10.3390/polym14163244

**Published:** 2022-08-09

**Authors:** Julija Volmajer Valh, Dimitrije Stopar, Ignacio Selaya Berodia, Alen Erjavec, Olivera Šauperl, Lidija Fras Zemljič

**Affiliations:** 1Institute of Engineering Materials and Design, Faculty of Mechanical Engineering, University of Maribor, Smetanova 17, SI-2000 Maribor, Slovenia; 2AKG–Robust Plastics Group, Jetfloat International GmbH, Oberer Markt 111, AT-8410 Wildon, Austria

**Keywords:** chitosan, active packaging, PET, recycling, reusability

## Abstract

Since millions of tons of packaging material cannot be recycled in conventional ways, most of it ends up in landfills or even dumped into the natural environment. The researched methods of chemical depolymerization therefore open a new perspective for the recycling of various PET materials, which are especially important for packaging. Food preservative packaging materials made from PET plastics are complex, and their wastes are often contaminated, so there are no sophisticated solutions for them in the recycling industry. After integrating the biopolymer chitosan, which is derived from natural chitin, as an active surface additive in PET materials, we discovered that it not only enriches the packaging material as a microbial inhibitor to reduce the bacteria *Staphylococcus aureus* and *Escherichia coli*, thus extending the shelf life of the contained food, but also enables economical chemical recycling by alkaline or neutral hydrolysis, which is an environmentally friendly process. Alkaline hydrolysis at a high temperature and pressure completely depolymerizes chitosan-coated PET packaging materials into pure terephthalic acid and charcoal. The products were characterized by Fourier-transform infrared spectroscopy, proton nuclear magnetic resonance spectroscopy, and elemental analysis. The resulting reusable material represents raw materials in chemical, plastic, textile, and other industries, in addition to the antimicrobial function and recyclability itself.

## 1. Introduction

In Europe, about 40% of plastic is used for packaging, which, in turn, becomes immediate waste [1]. The production and consumption of plastic materials in the food packaging industry increases from day to day [2], with polyethylene terephthalate (PET) being the most widely used due to its mechanical, chemical, and biochemical properties; transparency; flexibility; technical use; recyclability; and economical significance. It occurs in the form of sheets, films, fibers, composites, etc. [3]. It is estimated that more than 359 million tons of plastic is produced annually. More than a third of the total plastics produced are intended for the packaging sector. Most of the food packaging material is still landfilled, and a lot of it is even dumped in the environment [4,5]. Furthermore, food packaging is one of the most widespread applications, and its benefits have led to significant development and innovations in this area in recent years [6,7]. For example, in order to introduce the preservation from the PET food packaging, different additives, mostly synthetic ones, are used to prolong the shelf life of the products packaged. The integration (as fillers or coatings) of additives lead to the PET materials’ heterogeneity, resulting in more complex and contaminated waste [8]. With mechanical recycling being the most used, such mentioned waste is thus inappropriate and therefore subjected to the problematic landfilling.

Accordingly, there are two problems in the whole end-of-life analytical aspect of the PET food packaging: environmentally unfriendly preservation additives, used in PET coating, and the recyclability of complex PET packaging materials.

Keeping that in mind, a complete solution with a natural additive to the PET material that would equivalently enhance the preservation of food, possibly with an environmentally friendly recycling process that would fully degrade the complex material to its basic compounds, was the perfect milestone for our research. This growing demand for natural substances to be applied as activators in the development of active packaging is also strongly in accordance with an increased health and environmental awareness, as well as legislation requirements.

### 1.1. The Additives

In many applications, PET surfaces come into direct contact with food. The concept of active packaging refers to the incorporation of certain additives into the packaging film or within the packaging material bulk through extrusion or injection blow molding [9]. This is performed in order to ensure the perfect preservation of food, the extension of shelf life, and quality improvement, as well as the safety of the customer [10]. In active packaging, the enrichment of basic packaging materials with antimicrobial agents is trending. Microbial inhibition of the surface achieved by coating with functionalized additives ensures the safety of human health from the pathogenic microorganisms that compromise the ability of the packaging material to preserve the contained food.

The main functions of this type of packaging are microbicidal (reducing the infectivity of microbes) and microbiostatic (inhibiting the growth of microorganisms). Their purpose is to prolong a delay phase during microbial growth, thus reducing the microorganisms’ growth rate and inhibiting food aging.

Various antimicrobial agents may be incorporated [10] in the packaging or food system, including chemical antimicrobials; antioxidants; biotechnological products such as enzymes; antimicrobial polymers; and natural antimicrobials, such as extracts and gas. Antimicrobial agents can be classified into three groups: chemical agents, natural agents, and probiotics [11]. The main commercial products are still mostly based on synthetic and inorganic preservation promoting additives (e.g., chloride dioxide, benzoic anhydride, silver salts, etc. [9,11,12,13,14,15,16]).

In the search of an environmentally friendly, biodegradable, and natural substance, chitosan stands out in regard to possessing the mentioned properties. It is obtained from chitin shells of shrimp by partial deacetylation and has numerous biomedical applications. There are several publications that presented the use of chitosan macromolecular solutions or nanoparticles or chitosan mixtures with other polymers or polyphenols as edible films or coatings for foils [16,17,18,19,20,21,22,23]. To the best of our knowledge, such materials have not yet been studied from the point of view of end-of-life analysis. It must be noted that, after use, the packaging material represents the waste, together with the coated substrate. So, it is of a great meaning to also understand the influence of the coating on material degradability and recyclability. From the environmental point of view, it is an advantage to use biodegradable coatings such as chitosan, as it is hypothesized to be fully degradable and may, after chemical treatment, lead to the formation of charcoal. In addition, it is also a highly safe additive and is FDA approved [18].

### 1.2. Recycling

There are still large quantities of plastics that, instead of being repaired or reused, are dumped or landfilled after use [4,5]. The recycling management (Figure 1) suggests that the plastic material alone or modified should be repaired or reused at the end of the plastic’s life rather than becoming a residue.

Today, mechanical recycling is the most widely used type of plastics recycling [24,25]; however, not every plastic material is suitable for this type of recycling [26].

PET-based waste streams are suitable materials for treatment by mechanical processes aimed at recovering solid plastic waste for reuse [27]. Due to the degradation and heterogeneity of the solid plastic waste, only single polymer plastics can be mechanically processed, thus excluding all more complex and contaminated waste. Mechanically recycled products often end up being simply incinerated or disposed of in the field because of the poor quality of the recycled material. The use of mechanically recycled plastic, especially PET, in contact with food requires special attention. It may contain health-concerning degradation products of polymers and additives resulting from a barren mechanical recycling process, incidental harmful impurities from previous use and common consumer misuse, cross-contamination from waste disposal, and environmental contamination. These contaminants can migrate from packaging into food and represent a risk to human health. The mechanical recycling of food packaging waste into new food packaging therefore poses particular challenges, especially regarding health safety issues. Furthermore, food-packaging regulations in Europe and the USA require the same level of safety for chemicals that migrate into food, for all recycled and new materials alike [28,29,30,31]. In Europe, the use of recycled plastics in Food Contact Materials (FCMs) is specifically regulated under the Plastics Recycling Regulation (EC 282/2008) [32].

Under the European Directive [33] on the reduction of the impact of certain plastic products on the environment, beverage bottles made from polyethylene terephthalate as the main component (PET bottles) must contain at least 25% recycled plastic from 2025 and, from 2030, 30% recycled plastic.

### 1.3. Challenge

Mechanical recycling of plastic materials may display some short-term advantages for the environment, but as seen, long-term results may not be so good. Thus, the preferred method for recycling of plastic is to degrade it into basic components as a high-added-value substrate for chemical, pharmaceutical, and polymer industry. In this context, the advantage of chemical recycling over mechanical recycling is the isolation of the monomeric unit, which is a secondary raw material and is further used in the polymerization stage. Depolymerization, i.e., chemical recycling, seems to be an interesting way of recycling PET [34]. There are several processes of splitting the PET chains, such as glycolysis, methanolysis, hydrolysis, hydrogenation, gasification, pyrolysis, chemical depolymerization, photodegradation, ultrasonic decomposition, or decomposition in a microwave reactor. Nevertheless, mostly used are the first three approaches [35].

In the presented research paper, we combined the effective preservation ability of the environmentally friendly chitosan added to PET and the full chemical recycling of the obtained functionalized packaging material, i.e., those coated by chitosan macromolecular solution. After a successful neutral and alkaline hydrolysis and isolation of monomer units, we achieved a complete depolymerization of the functional packaging material to ethylene glycol, terephthalic acid; both from PET origin and bio/charcoal as flow from chitosan coating, which as resalable materials presented another added value beside the antimicrobial function and recyclability itself. PET food packaging waste can be considered as a source of important secondary raw materials. In this way, isolated terephthalic acid presented secondary raw material and can replace fossil terephthalic acid used in the synthesis of PET materials. From the other side, many publications have indicated that biochar can increase soil fertility, increase agricultural productivity, and provide protection against some foliar and soil-borne diseases. Furthermore, adding biochar can prolongate the time of cultivation, and it may reduce pressure on forest areas to agriculture [36]. So, the chemical recycled chitosan as PET coating may also find some of this agriculture application.

## 2. Materials and Methods

### 2.1. Materials and Reagents

PET amorphous foil with a thickness of 350 µm, produced by Goodfellow, was used and further treated by CO_2_ plasma within a discharge chamber made of a Pyrex cylinder, with a length of 0.6 m and an inner diameter of 0.036 m, as in the details given in Reference [16].

Other chemicals used in the experimental work were supplied by Sigma-Aldrich (chitosan and terephthalic acid) or by GRAM -MOL, Zagreb (NaOH and 37% HCl). A 0.53 M solution of NaOH and chitosan with a concentration of 1% (*w*/*w*) was prepared in double-distilled water. The pH value of the chitosan solution was adjusted to 3.6 by stirring with 37% HCl.

### 2.2. PET’s Coating with Chitosan

PET foil that was coated by using chitosan macromolecular solutions (PET–CHT) was prepared according to the following procedure. PET foils that were previously activated by CO_2_ plasma (20 s) in order to improve chitosan adhesion were cut into strips of 100 × 100 mm and cleaned in a bath with pure ethanol and sonicated with ultrasound (Transsonic 825/H) for 30 min. They were then immersed in double-distilled water and air-dried to a constant weight. Then 12.5 g of PET foil was added to the 1% (*w*/*w*) chitosan solution for 72 h. Finally, the foils were dried in a vacuum dryer (Kambič, vs. −25 °C) at 50 °C for 24 h.

### 2.3. PET Hydrolysis

#### 2.3.1. Neutral Hydrolysis

The water-based PET hydrolysis was carried out in a 1 L stainless-steel high-pressure reactor from Ecom, Slovenija. The experiments were performed with 12.5 g PET foil or with PET foil coated with chitosan (PET–CHT) in 250 mL deionized water at 250 °C for 10 min. The autogenous pressure achieved in the reaction vessel depends on the temperature, and depending on the amount of water used, it was in the range of (38 to 40) bar in the abovementioned experiments. The heating time required to achieve the desired reaction conditions was not considered in the hydrolysis reaction time; that is, t = 10 min means that the PET hydrolysis was stopped 10 min after reaching steady state. At the end of the process, the liquid phase was separated from the solid phase by filtration. The solid phase was dried, weighed, and characterized. Three repetitions of neutral hydrolysis were performed. The experimental conditions were based on our previous investigations on PET fibers [37].

#### 2.3.2. Alkaline Hydrolysis

The alkaline hydrolysis of PET foil and PET foil coated with chitosan (PET–CHT) were carried out in a 1 L stainless-steel high-pressure reactor from Ecom, Slovenija. The experiments were performed with 12.5 g PET foil in 250 mL in 0.53 M NaOH solution at 250 °C, autogenous pressure in the range of (38 to 40) bar, and a reaction time of 10 min after reaching the steady state. At the end of the process, the liquid phase was separated from the solid impurities by filtration. The stoichiometric amount (10.5 mL) of concentrated hydrochloric acid was added to the liquid phase, and the precipitated white powder was filtered, dried, weighed, and characterized. Three repetitions of the alkaline hydrolysis were performed.

### 2.4. Analytical Methods

The ATR–FTIR spectra were recorded with a Perkin Elmer Spectrum GX spectrometer. The ATR accessories (supplied by Specac Ltd., Orpington, UK) contained a diamond crystal. For each sample, a total of 16 scans were performed with a resolution of 4 cm^−1^; the depth analysis was 0.75 mm. All spectra were recorded at ambient temperature over a wavelength interval between 4000 and 650 cm^−1^. An elemental analyses of total carbon, hydrogen, nitrogen, and sulfur (CHNS analysis) was performed on Instrument Elementar vario Macro CHNS/CHN.

### 2.5. Antimicrobial Testing

Microbiological tests on coated PET foils, with a focus on the inhibitory effect of chitosan, were carried out according to the method of ASTM E 2149-01 (2002). To determine the reduction rate, the foils were exposed to three different type strains: *Staphylococcus aureus* ATCC 25923, *Escherichia coli* ATCC 25,922, and *Candida glabrata* EC 87069. The microorganisms were incubated as described in ASTM E 2149-01 (2002), except for *Candida glabrata*, which was incubated for two days at 25 °C.

The standard procedure was carried out in several stages:-Preparation of microbiological media (TSB, NA, and PCA),-Production of buffer solutions (PBS) and cleavage of bacteria,-Incubation (at 37 °C and 25 °C),-Evaluation of antimicrobial effectiveness.

The growth media used in the standard procedure were prepared in accordance with the procedure for the production of culture media. One sterile 250 mL Erlenmeyer flask with screw cap was used for each functionalized and non-functionalized slide, and one “inoculum only” sample was used for the current series. Then 50 ± 0.5 mL working dilution of the bacterial inoculum was added to each flask, and 0.9 g of a foil in a film was aseptically transferred to the bacterial suspension. The initial number of bacteria in suspension at time “0” was then determined by serial dilutions and the standard colony counting technique (ISO 4833, 2003) from the “inoculum only” flask. The number of bacteria determined at 1 min actually represents the initial concentration (0) of bacteria in the bacteria suspension. The test and control samples were placed in their individual flasks. Series of flasks were shaken on the wrist shaker, with maximum stroke, for 1 h ± 5 min. The number of bacteria in each sample was determined by colony count (ISO 4833, 2003) with NA and incubation of agar plates for 24 h at 37 °C, or at 25 °C (*C. glabrata*). Afterward, the grown colonies were counted by using the colony counting technique (ISO 4833, 2003), and the numbers were converted into the average colony forming units per milliliter (CFU/mL). This was followed by the calculation of the bacterial reduction as follows:(1)$$R=\frac{N_{1\;hour}-N_{1\;min}}{N_{1\;hour}}\times 100\left[\%\right]$$
where *R* is the bacterial reduction (%); *N*_1_ *_min_* is the number of bacterial colonies (CFU/mL) after 1 min incubation of test suspension; and *N*_1 *hour*_ is the number of bacterial colonies (CFU/mL) after 1 h of incubation of test suspension.

## 3. Results and Discussion

### 3.1. PET Coating and Its Active Concept Monitoring

The adsorption of the chitosan coating onto PET was confirmed by ATR–FTIR spectroscopy. Figure 2 shows the ATR–FTIR spectra of PET raw material foil (PET) and commercially available chitosan and chitosan functionalized PET (PET–CHT) foils previously plasma activated. The results show that chitosan was successfully distributed on PET surfaces, which is demonstrated by the presence of an -NH stretching peak in the range between 3300 and 3020 cm^−1^. With the appearance of the amino peak, the -CH stretch peak in the range between 2900 and 3800 cm^−1^ increased. The ATR–FTIR spectrum of functionalized PET foil by chitosan showed typical chitosan signals at 1647 and 1559 cm^−1^ (corresponding to the carbonyl stretching vibration (amide I) and the N-H bending vibration (amide II) of a primary amino group) and typical PET foil signals at 1714 cm^−1^ (corresponding to the carbonyl C=O stretching of ester bonds).

### 3.2. Antimicrobial Testing

The results in Table 1 show that functionalized foils mainly reduce the growth of Gram-negative bacteria, slightly less Gram-positive bacteria, as well as fungi that are also, to some extent, affected by chitosan coatings. It may be seen, through the use of this test, that foil PET–CHT expresses effective microbial inhibition. The successful reduction of both bacteria, *Staphylococcus aureus* and *Escherichia coli*, is extremely important, whilst these two bacteria are frequently responsible for the perishability of food in the packaging system. Bacterial attacks can occur during food processing, distribution, and sale and especially by inappropriate storage practices.

*E. coli* causes diarrheal diseases via the production of enterotoxin or cytotoxin, or plasmid-mediated virulence factors. If food is contaminated with Staph, the bacteria can multiply in the food and produce toxins that can make people ill [38] due to intestinal and gastric infection. Thus, it is of great importance in regard to food safety to develop the packaging to inhibit both of these bacteria. There is also some fungi inhibition present which is extremely important against mold and yeast associated with bread, vegetables and cheese [39].

### 3.3. Chemical Recycling—Isolation of Secondary Raw Material TPA

A neutral and alkaline hydrolysis of non-activated and activated PET foils was performed. From the diagrams in Figure 3 and Figure 4, it can be concluded that both hydrolysis reactions were performed almost equally, since the addition of 1% chitosan solution does not seem to change the process.

#### 3.3.1. Neutral Hydrolysis Condition

PET foil and PET–CHT foil were depolymerized by neutral hydrolysis at a high temperature and high pressure. From a theoretical point of view (Figure 5), the complete depolymerization of PET leads to two monomeric units, namely terephthalic acid (TPA) and ethylene glycol (EG). In the case of incomplete depolymerization, the liquid-phase PET consists mainly of water, ethylene glycol, and diethylene glycol, and the solid phase consists mainly of untreated PET oligomers and TPA.

In the study, we focused mainly on the solid fraction. The mass of the isolated solid fraction after neutral hydrolysis of PET foils and PET chitosan-coated foils are shown in Table 2 and Table 3. It was difficult to accurately measure the amounts of solid products obtained after each neutral hydrolysis because of unavoidable weight losses during product processing. The isolated solid fraction after neutral hydrolysis of PET foil was white and homogeneous, and from the coated PET foil, the white chitosan was heterogeneous, with some black particles. As we expected, the chitosan coating affects the purity of the isolated solid product.

The ATR–FTIR spectra of the isolated solid fraction after neutral hydrolysis of PET foil and PET chitosan-coated foil were recorded and compared with commercially available pure TPA. Commercially available TPA had the following characteristic bands specific for aromatic dicarboxylic acids at 3063 cm^−1^ (O-H), 1675 cm^−1^ (C=O), and 1280 cm^−1^ (C-O) and specific for a 1,4-disubstituted benzene ring at 1137 and 1018 cm^−1^ (Figure 6, green spectrum).

While the ATR–FTIR measurement of solid product obtained after neutral hydrolysis of PET foil (blue spectrum) indicates the presence only of TPA, in the ATR–FTIR spectrum of the solid product obtained after neutral hydrolysis of PET foil coated with chitosan (pink spectrum), some new signals were detected at 3460 cm^−1^ that could be associated with the presence of chitosan itself or a product from the incomplete depolymerization of PET. The insignificant peak at 1716 cm^−1^, which overlapped with the peak of the carboxylic acid at 1685 cm^−1^ attributed to the ester bond, is due to incomplete depolymerization of the PET foil according to our previous results. To obtain pure recycled TPA without residual oligomers, the solid product must be cleaned from PET chitosan-coated foils after neutral hydrolysis. As a cleaning step the method of base–acid precipitation was tested. The solid product after the neutral hydrolysis of the chitosan-coated PET foil was placed in 1.04 M sodium hydroxide to form the sodium salt of TPA. Insoluble products were filtered off. In the next step, the stoichiometric amount of concentrated hydrochloric acid was added, and the precipitated white powder was filtered, dried, and characterized by ATR–FTIR spectroscopy.

Figure 7 shows the ATR–FTIR spectra of commercially available pure TPA (B—blue spectrum), an isolated solid fraction after neutral hydrolysis of chitosan-coated PET foil (A—black spectrum), and the purified product (C—green spectrum). From the ATR–FTIR spectra, it can be concluded that the purification was successful. The ATR–FTIR spectrum of purified product is identical to the ATR–FTIR spectrum of commercially available pure TPA.

However, in the case of PET chitosan-coated foil, neutral hydrolysis is not sufficient, and an additional purification step is needed. This process not only prolongs the recycling process of PET chitosan-coated foils, but also makes it more expensive (more chemical compounds and chemical processes are needed), and the final quantity and purity of TPA is lower because a certain amount of material is lost at each step. In order to avoid the purification step after neutral hydrolysis, we further performed alkaline hydrolysis of PET chitosan-coated foil.

#### 3.3.2. Alkaline Hydrolysis Conditions

PET chitosan-coated foil was depolymerized by alkaline hydrolysis at high temperatures and high pressure (Figure 8). Under neutral hydrolysis conditions, all impurities already present in the polymer material are converted into TPA, which is why numerous purification steps are usually required to achieve the desired TPA purity. With alkaline hydrolysis, the purification step is not necessary. The alkaline hydrolysis of PET leads to the formation of a liquid product consisting of TPA in the form of sodium salt and impurities insoluble in water. The impurities were filtered off, and by adding acid, the dissociated sodium salt of TPA is transformed into solid TPA, which precipitates from the solution.

The mass of the isolated product after alkaline hydrolysis of PET foils coated with chitosan and after addition of acid is shown in Table 4.

The isolated product after alkaline hydrolysis was characterized by ATR–FTIR spectroscopy. From the spectra in Figure 9, we can conclude that the product obtained after alkaline hydrolysis (black spectrum) is identical to the ATR–FTIR spectrum of commercially available TPA (blue spectrum).

### 3.4. Characterization of Impurities

Impurities isolated after the purification step of neutral hydrolysis and after alkaline hydrolysis were characterized by ATR–FTIR spectroscopy (Figure 10) and by CHNS analysis. From the ATR–FTIR analysis, we can conclude that the impurities do not contain typical organic functional groups, since we do not see any intense signals in the spectra. From this, we can conclude that the impurities are from burnt chitosan and PET additives. Using CHNS analysis, the high carbon content (66%) in the isolated impurities was determined.

According to the European Biochar Certificate (EBC), isolated impurities must meet the following criteria: the carbon content of the biochar must be higher than 50% of the dry mass, the H/C_org_ molar ratio must be less than 0.7, and the O/C_org_ molar ratio must be less than 0.4. Measurements performed on isolated impurities showed the following results: the carbon content is 66%, H/C_org_ molar ratio is 0.057, and O/C_org_ molar ratio is 0.16, which prove that biochar is at stake.

## 4. Comparison of the Classical TPA Synthesis Route with the Chemical Recycling of PET Food Packaging Waste and Isolation of TPA

Terephthalic acid (TPA) is a major component in the polyester industry and is used in the production of polyester terephthalate (PET) and polyester fibers. Today, TPA is a synthesis from the non-renewable resources from the oxidation of para-xylene by the commercialized AMOCO process, which uses a homogeneous catalyst of cobalt and manganese together with a corrosive bromine compound as promoter. This process is carried out in acidic medium at a high temperature (175–225 °C) [41].

Traditionally, para-xylene is produced by the catalytic reforming of various crude oil streams to produce a mixture of benzene, toluene and ortho-xylene, meta-xylene, and para-xylene [42,43]. High-purity para-xylene is required for the synthesis of TPA, and several technologies have been developed to purify and maximize para-xylene yields.

High-quality terephthalic acid is required for the synthesis of PET polymers. Therefore, the production of high-quality terephthalic acid with an impurity content of 4-carboxybenzaldehyde of less than 25 ppm [44] has been a major problem in the past. Before the commercialization of the AMOCO process to produce high-quality terephthalic acid in the late 1970s, extensive research and studies were carried out over several years. However, in addition to the AMOCO process, other catalytic processes to produce terephthalic acid by direct oxidation have been extensively studied to find the best process for terephthalic acid production for industrial purposes. Researchers are still working on this topic and are looking for the best options. The classical linear concept of PET packaging preparation is presented in Figure 11. After use, it is disposed of in landfills.

With the chemical recycling of PET material, we want to move to a closed-loop recycling concept, as shown in Figure 12. TPA can be isolated from PET waste, which is usually disposed of in landfills, with high negative environmental impacts, instead of being produced from non-renewable resources. The process is new and mainly connected with the laboratory scale. Some projects, such as Resyntex [45] and Demeto [46], are opening new avenues in the industrial pilot production of TPA from PET waste.

Some companies have also already started to recycle on an upward cyclical basis. The Japanese company Teijin Fibers Ltd., for example, has developed an upcyclical recycling process for recycling PET bottles and PET fibers based on the chemical process of methanolysis. The recycling is carried out at high temperatures and high pressures in methanol, the main products obtained being dimethyl terephthalate (DMT) and ethylene glycol (EG) [47]. Like all processes, the methanolysis process has advantages and disadvantages. The main advantage is that the DMT obtained after recycling is of the same quality as new DMT. The main disadvantage is the high cost associated with the separation and refining of the mixture of reaction products (glycols, alcohols, and phthalate derivatives); contamination of such mixtures with water can form different azeotropes [48]. A further disadvantage is that DMT must be converted into TPA, whereas, today, in all PET production processes, terephthalic acid (TPA) is used as the raw material instead of DMT, thus making the methanolysis process considerably more expensive. The upcycling recycling process looks very optimistic, but some optimization still needs to be performed soon. We need to be aware that waste can be considered as a source of raw material.

## 5. Conclusions

We found that the chitosan coatings act as an antimicrobial agent and prolong the shelf life of the packaged food products. From the chemical recycling point of view, this coating does not affect the quality of the isolated secondary raw material when we add the purification steps of the isolated monomeric unit TPA or when recycling is carried out under alkaline conditions. Moreover, two streams of pure raw materials, namely terephthalic acid and charcoal, were obtained. Terephthalic acid is a valued starting component in the chemical, pharmaceutical, and plastic industries. Activated carbon obtained from chitosan can be used as a cleaning sorbent or as an additive to soils for better soil fertility.

The advantage of chemical recycling is that we obtain the basic raw material compounds. With mechanical recycling, the mechanical properties of the material change, and recycling PET cannot be replaced 100%.

With this research, we are providing a complete solution to the packaging industry: an environmentally friendly functionalization of a food packaging material and its economical and full recyclability.

## Figures and Tables

**Figure 1 polymers-14-03244-f001:**
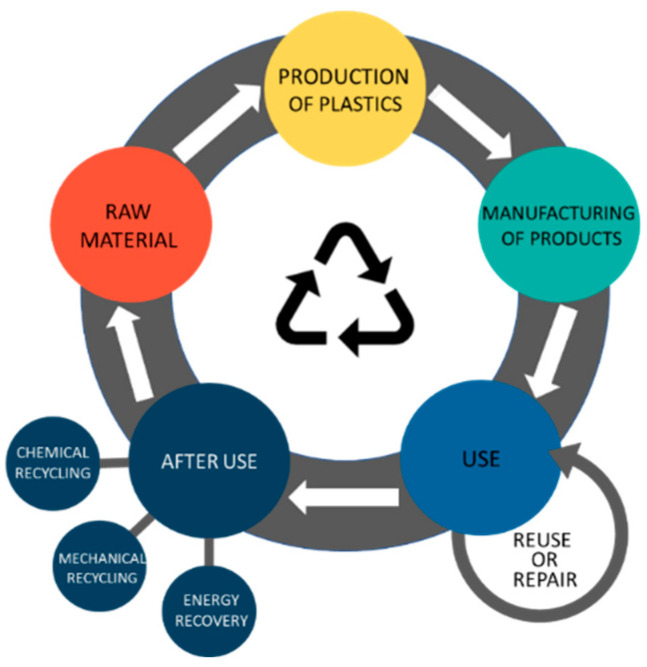
Recycling management concept.

**Figure 2 polymers-14-03244-f002:**
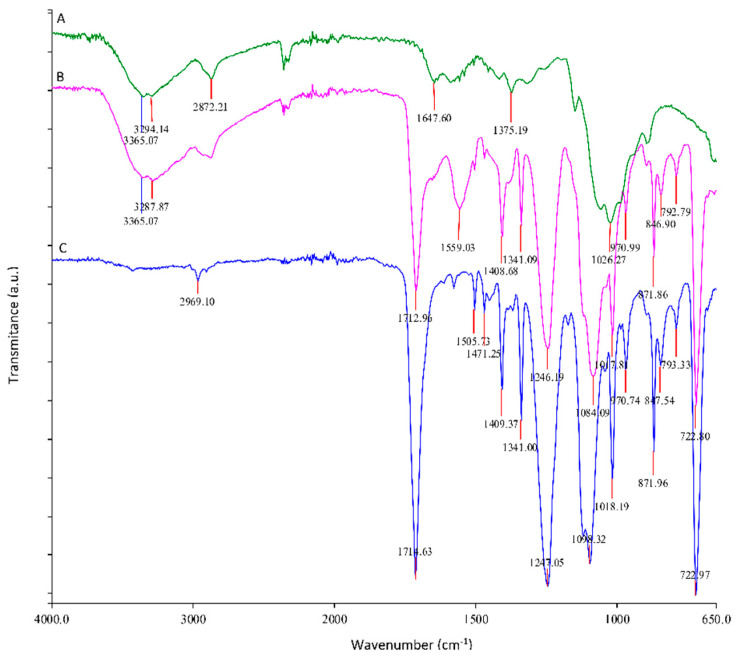
ATR–FTIR spectra of PET foil (C spectrum), commercially available chitosan (A spectrum), and functionalized PET foil by chitosan (PET–CHT) (B spectrum).

**Figure 3 polymers-14-03244-f003:**
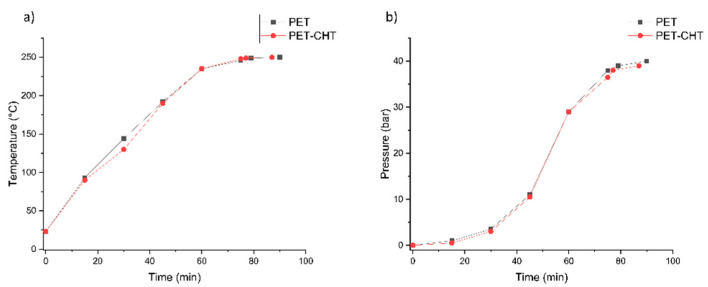
(**a**) Time–temperature dependence neutral hydrolysis of PET and PET–CHT. (**b**) Time–pressure dependence neutral hydrolysis of PET and PET–CHT.

**Figure 4 polymers-14-03244-f004:**
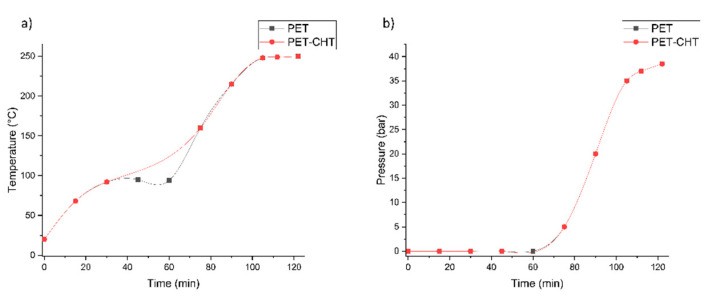
(**a**) Time–temperature dependence alkaline hydrolysis of PET and PET–CHT. (**b**) Time–pressure dependence alkaline hydrolysis of PET and PET–CHT.

**Figure 5 polymers-14-03244-f005:**
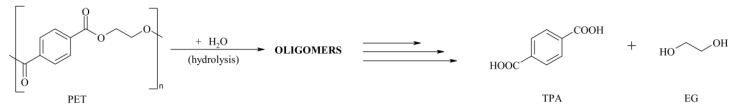
Neutral hydrolysis of PET [40].

**Figure 6 polymers-14-03244-f006:**
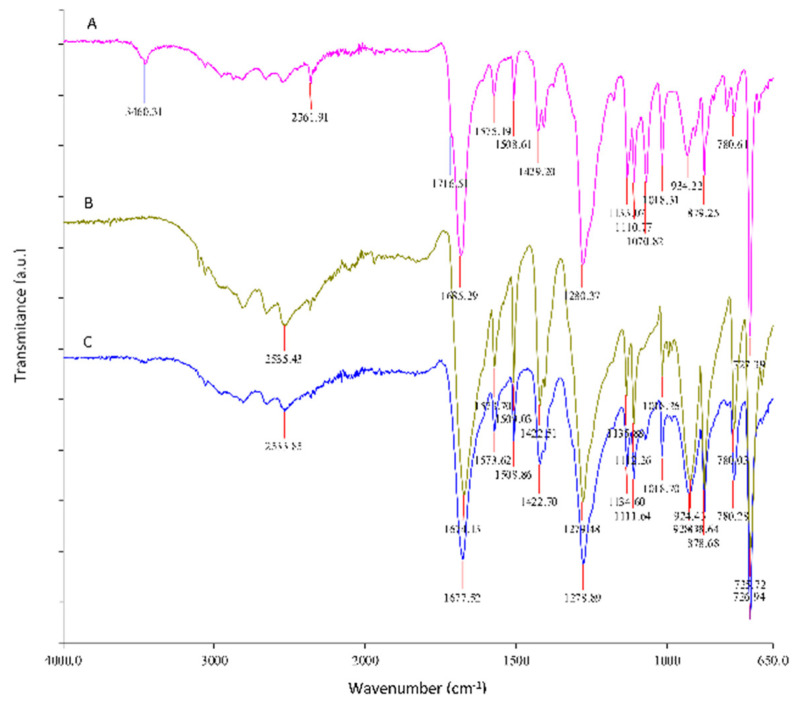
ATR–FTIR spectra of commercially available pure TPA (B–green spectrum), isolated solid fraction after neutral hydrolysis of PET foil (C—blue spectrum), and isolated solid fraction after neutral hydrolysis of PET foil coated with chitosan (A—pink spectrum).

**Figure 7 polymers-14-03244-f007:**
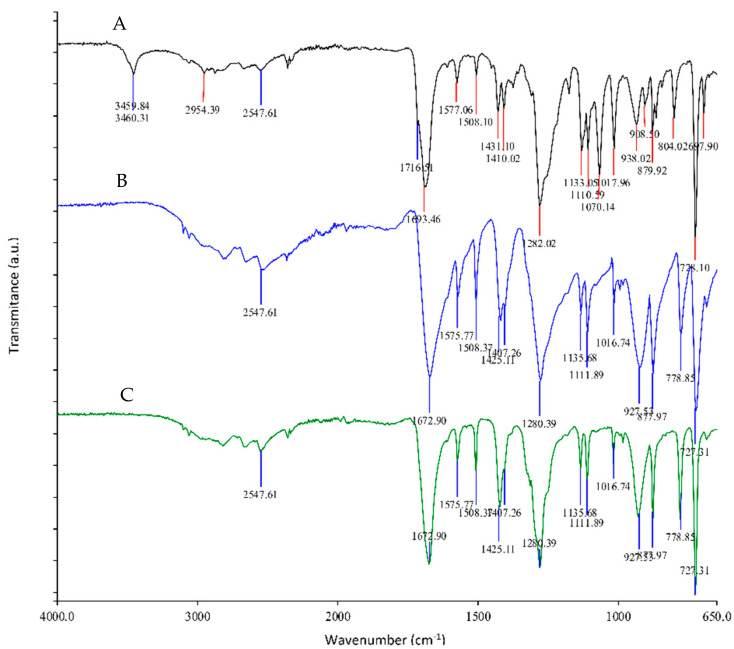
ATR–FTIR spectra of commercially available pure TPA (B spectrum), isolated solid fraction after neutral hydrolysis of PET foil coated with chitosan (A spectrum), and purified product (C spectrum).

**Figure 8 polymers-14-03244-f008:**
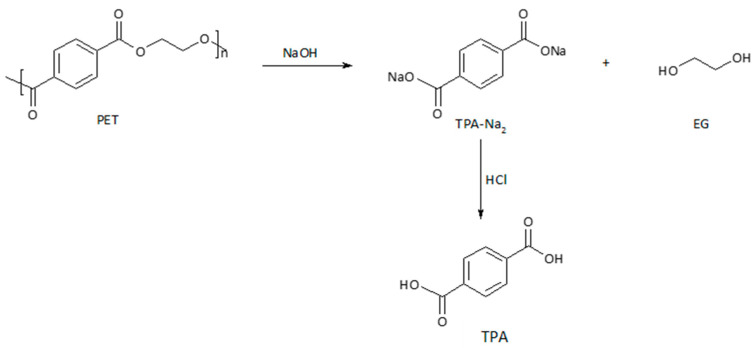
Alkaline hydrolysis of PET [40].

**Figure 9 polymers-14-03244-f009:**
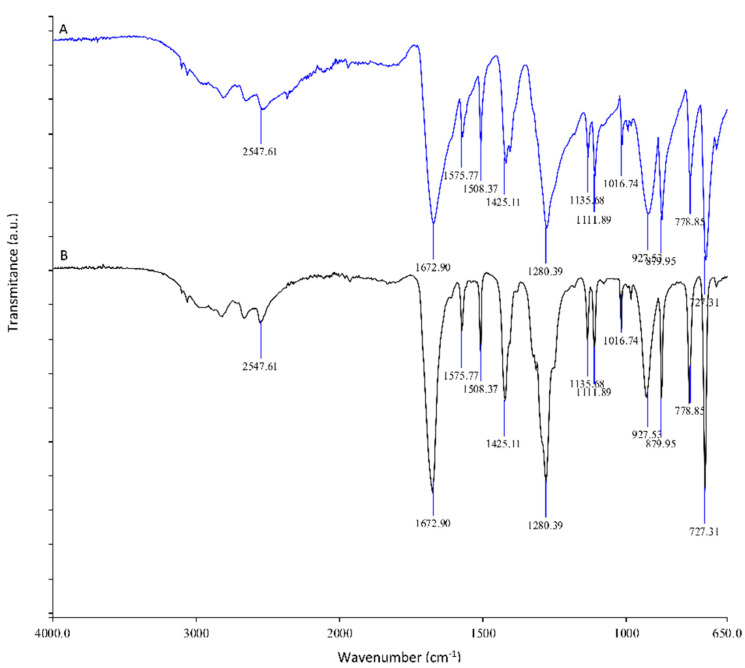
ATR–FTIR spectra of commercially available pure TPA (A spectrum), isolated solid fraction after alkaline hydrolysis, and addition of acid of PET foil coated with chitosan (B spectrum).

**Figure 10 polymers-14-03244-f010:**
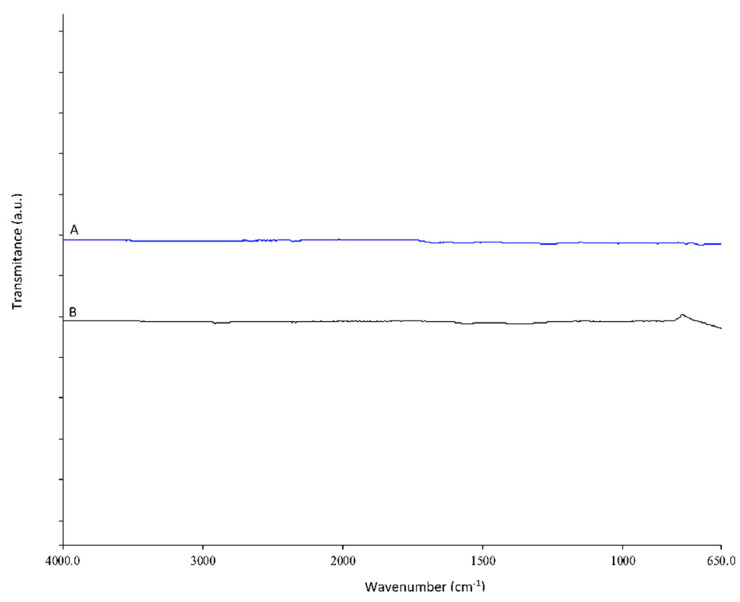
ATR–FTIR spectra of impurities obtain after purification step from neutral hydrolysis (A-blue spectrum) and after alkaline hydrolysis (B-back spectrum).

**Figure 11 polymers-14-03244-f011:**
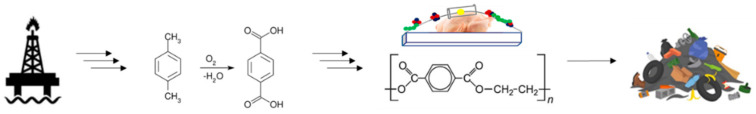
Linear concept.

**Figure 12 polymers-14-03244-f012:**
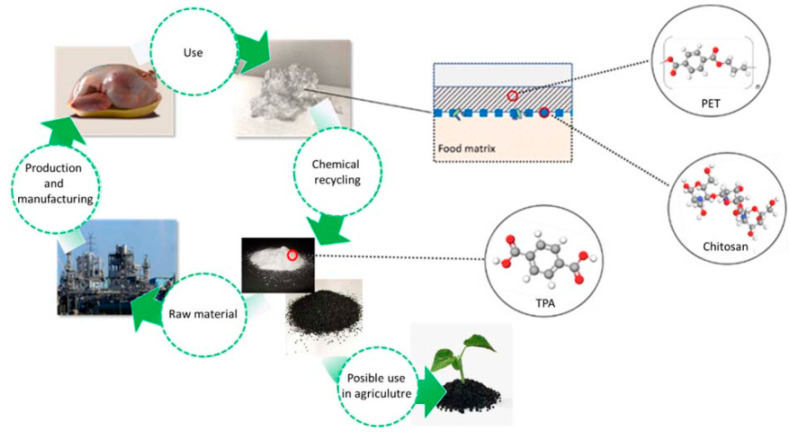
Circular-economy concept.

**Table 1 polymers-14-03244-t001:** Reduction, R (%), of selected microorganisms after exposure to differently treated PET foil, as obtained by ASTM E 2149-01 (2002) standard test.

SAMPLES	Microbial Reduction (%)		
	*Staphylococcus aureus* ATCC 25923	*Escherichia coli* ATCC 25922	*Candida glabrata* EC 8706
PET foil	0 ± 0	17 ± 1	0 ± 0
PET–CHT	76 ± 2	91 ± 2	42 ± 2

**Table 2 polymers-14-03244-t002:** Mass of isolated solid fraction after neutral hydrolysis of PET foil.

Experiment #	m/(g)
1	8.6
2	8.7
3	8.9

**Table 3 polymers-14-03244-t003:** Mass of isolated solid fraction after neutral hydrolysis of PET foil coated with chitosan.

Experiment #	m/(g)
1	9.2
2	9.0
3	9.2

**Table 4 polymers-14-03244-t004:** Mass of isolated solid fraction after alkaline hydrolysis of PET foil coated with chitosan and after the addition of acid.

Experiment #	m/(g)
1	10.2
2	9.9
3	10.0

## Data Availability

The data presented in this study are available on request from the corresponding author.

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
