# Peer review of "Economical Chemical Recycling of Complex PET Waste in the Form of Active Packaging Material"

_polymers, 2022, doi:10.3390/polym14163244_

Round 1
Reviewer 1 Report
The research work is focused on one of the most used materials in food packaging, polyethylene terephthalate (PET). Specifically, the authors demonstrate that the coating of PET containers by depositing a chitosan film improves the antimicrobial functionality of the packaging and favors its chemical recycling at the end of its life through hydrolysis, especially in alkaline conditions. This recycling procedure appears to be an economical method for the production of terephthalic acid, widely used as a raw material for the synthesis of polyesters.
Despite the scientific relevance and environmental sustainability of the aspects covered, the manuscript must be reread carefully to eliminate both typos and considerations not pertinent to the description of the activities carried out.
In detail, some points to be changed are indicated below.
First of all, the title of the paper is not appropriate and should be changed. The authors write "crustacean inspired food preservation" but this concept is not mentioned in the text.
Line 84: The sentence starting with "Were until now ..." is incomplete. Please correct.
Lines 95-103: Guidelines for writing contributions are incorrectly indicated. Please delete
Row 159: the thickness of PET Amorphous foil is reported in mm. Obviously this is a mistake.
Line 179: Delete the word "and" after "with".
Section 2.4: I would suggest renaming this section as "Spectroscopic analysis".
Line 204: what is meant by CHNS analysis? The abbreviations must be made explicit at least the first time to allow the understanding of the contents even to readers less familiar with these techniques.
Line 259: Correct "extend" to "extent".
Sections 3.3.1 and 3.3.2 titles: I would suggest deleting the word "Under".
Lines 286-287: the sentence is convoluted. Please rephrase.
For the above, minor revisions are necessary to improve the quality and, therefore, the added value of the research.
Reviewer 2 Report
In this study, the biopolymer chitosan was integrating in the PET as an active surface additive which extends the shelf life of the contained food, but also enables economical chemical recycling by alkaline or neutral hydrolysis, which is an environmentally friendly process. This work is very innovative and the experimental design and results are convincing. I think it can be publish in polymers after minor revise,
(1) Line. 41 Reference error.
(2) I suggest that all infrared spectrograms figures be redesigned and modified to improve resolution and aesthetics.
(3) Line 160,169. CO2
(4) Figure 3, 4 needs to a high resolution one.
(5) Fig. 5 and 8 need reference.
(6) Some language problems need to be revised.
